# Preadapted to adapt: underpinnings of adaptive plasticity revealed by the downy brome genome

Samuel R. Revolinski [1], Peter J. Maughan[2], Craig E. Coleman[2] & Ian C. Burke [1✉]

*Bromus tectorum* L. is arguably the most successful invasive weed in the world. It has fundamentally altered arid ecosystems of the western United States, where it now found on an excess of 20 million hectares. Invasion success is related to avoidance of abiotic stress and human management. Early flowering is a heritable trait utilized by *B. tectorum*, enabling the species to temporally monopolize limited resources and outcompete the native plant community. Thus, understanding the genetic underpinning of flowering time is critical for the design of integrated management strategies. To study flowering time traits in *B. tectorum*, we assembled a chromosome scale reference genome for *B. tectorum*. To assess the utility of the assembled genome, 121 diverse *B. tectorum* accessions are phenotyped and subjected to a genome wide association study (GWAS). Candidate genes, representing homologs of genes that have been previously associated with plant height or flowering phenology traits in related species are located near QTLs we identified. This study uses a high-resolution GWAS to identify reproductive phenology genes in a weedy species and represents a considerable step forward in understanding the mechanisms underlying genetic plasticity in one of the most successful invasive weed species.

[1] Department of Crop and Soil Sciences, Washington State University, Pullman, WA, USA. [2] Department of Plant & Wildlife Science, Brigham Young University, Provo, UT, USA. ✉email: icburke@wsu.edu

**B**romus tectorum is the most abundant invasive weed in North America. In the western United States, it is estimated to infest 31.4% (210,000 km$^2$) of the Great Basin[1]. It is most notorious for and especially problematic in non-cropland and rangelands of the intermountain west where *B. tectorum* invasion has altered intervals between fires from at least 60 years to less than 5 years[2,3], exacerbating the degradation of ecosystems caused by climate change[1,4]. *Bromus tectorum* is also a damaging weed in agricultural crops, causing substantial yield loss in winter wheat (*Triticum aestivum* L.) across a large proportion of the western North American wheat producing region[5].

The history of *Bromus tectorum* (L.) in North America is proposed to have begun as the arrival of a small number of founder genotypes from multiple introductions originating from a wide range of native habitats in Eurasia[6]. Due to multiple introductions of *B. tectorum* from Eurasia to North America the overall genetic diversity in Eurasia is higher than in North America but within populations the genetic diversity was higher in North America because the North American populations of *B. tectorum* are often a heterogenous mix of the introduced genotypes[6]. The seed likely arrived in animal bedding, grain contaminants, or it was imported intentionally as a potential forage[7]. *Bromus tectorum*, known as cheatgrass or downy brome, is predominantly a self-fertilizing, cleistogamous species, adapted to multiple ecosystems in its native range[8]. Although the species is distributed across North American ecosystems, the population has retained genetic signatures that trace back to ancestral populations and ecosystems[6]. Thus, the success of *B. tectorum* invasiveness is due in large part to the diversity of these original populations that exhibit substantial levels of plasticity in phenological traits. Plasticity allows individual *B. tectorum* plants to establish themselves in a locale by being pre-adapted to local changes in the availability of in-season resources[9], while populations of *B. tectorum* are composed of an assemblage of diverse genotypes, facilitating success in response to long term and local variation in climate[10–12]. Such adaptive variation in life cycle traits[13,14] hinders management efforts in both natural and agricultural ecosystems. In short, *B. tectorum* individuals and populations express adaptive plasticity and, when expressed as earlier and variable reproductive phenology, allow *B. tectorum* to outcompete for limited resources at the expense of the native plant community[9]. Where it successfully invades, *B. tectorum* germinates and flowers early, facilitating access to limited resources, usually moisture, well before the native vegetation or crops can compete successfully for these resources[15,16].

Phenotypic variation in *B. tectorum* for adaptive traits like the aforementioned flowering time, but also seed dormancy and vernalization are undoubtedly major drivers of *B. tectorum's* highly successful invasion across a wide variety of North American ecosystems[13,15–17]. In model and crop *Poaceae* species, growth and flowering phenology is controlled by an array of light and temperature sensitive gene pathways. Unfortunately, little is currently known about the genetic basis of adaptive variation in *B. tectorum*. Orthologous genes, such as *FT* (flowering time), *VRN1* (*VERNALIZATION1*), and *VRN2* (*VERNALIZATION2*) are likely contributing factors[18]. As a member of the Pooideae subfamily in the Poaceae family, *B. tectorum* is closely related to the Triticeae tribe, which includes the agriculturally important species barley (*Hordeum vulgare* L.), wheat (*Triticum aestivum* L.) and rye (*Secale cereale* L.)[19,20]. Thus, extensive genomic resources, including well annotated genomes, in these sister taxa are available that will facilitate comparative genomics and gene discovery efforts in *B. tectorum*.

Here we report a high-quality, annotated reference genome for *B. tectorum*. We utilize the reference genome in a genome wide association study (GWAS) to identify candidate genes for reproductive phenology (days to first joint, days to first visible panicle, days to first ripe seed and number of tillers) and plant height—traits that are known to directly influence the success of *B. tectorum* as an invasive weed species. Genome wide association studies (GWAS) are particularly useful for dissecting complex traits in species where controlled crossing is not practical or possible[21–25]. Our study demonstrates a successful application of GWAS for the identification of QTL controlling heritable reproductive phenology traits in a weedy, highly invasive species. The identification of candidate gene targets controlling important climate and management adaptive characteristics underlying these QTL suggests how *B. tectorum* might respond to climate change, thus enabling the development enhanced and more reliable management practices for this highly invasive and problematic weed species.

## Results and discussion

**Genome assembly**. The Omni-C chromosomal assembly produced a near complete assembly for *B. tectorum*. The assembly was 2,482 megabases (Mb) in total length (Supplementary Table S1) with 1298 genes and 82% of the genome consisting of repetitive elements (Supplementary Table S2). The contig N50 was 19.4 Mb, the scaffold N50 was 357.4, and the resulting assembly contained 92.1% of the BUSCO genes with 259 gaps. The final assembly had a L50 of four and a L90 of seven corresponding to the seven chromosomes ($x = 7$) expected for members of the Pooideae in the Poeae, Aveneae, Bromeae and Triticeae tribes[26].

Based on CDS comparisons using MCScanX[27], the seven largest scaffolds of the *B. tectorum* reference genome were found to have a one-to-one syntenic relationship with the seven chromosomes of the barley genome—not surprising given their phylogenetic proximity. As expected, synteny between *B. tectorum* and *H. vulgare* was highest in the chromosome arms where gene density is known to be high and lowest though the centromeric region where gene density is substantially reduced (Fig. 1). A translocation is present between chromosomes two and five where the first half of *Bt5* has synteny with chromosome two in barley and the second half of *Bt2* has synteny with chromosome 5 in barley (Fig. 1).

**Phenotypic analysis**. Broad Sense Heritability (Reliability), the mean, standard deviation, min, max, and correlation of Best Linear Unbiased Estimates (BLUEs) were calculated to characterize the variation in traits and obtain a measure of the effect of each genotype. The distributions of BLUEs for reproductive phenology traits indicated a bimodal distribution (Fig. 2), where more rapidly flowering and taller plants consisted of accessions from Washington and GRIN collections. In contrast, Montana accessions flowered later and were shorter (Supplementary Data 1, Supplementary Data 2 and Fig. S1). Broad sense heritability was high for all the traits and ranged from 0.94 for tiller number to 0.99 for days to first visible panicle (VPN) (Table 1). Genotypes had a wide range of BLUEs for all traits measured: plant height (PH) ranged from 36.5 to 89 cm, number of tillers ranged from 5.5 to 38.5, and days to first ripe seed (FRS) ranged from 43.2 to 112.5 d (Table 1). Spearman correlations between BLUEs of different reproductive phenology traits were all above 0.95 (Fig. 2). Reproductive phenology and PH traits were moderately negatively correlated, with Spearman correlations ranging from the −0.44 (PH and FRS) to −0.53 (PH and J1).

**Genome-wide association mapping for height, tiller number, and phenology traits**. To identify regions of the genome associated with variation in adaptive traits, a GWAS was performed

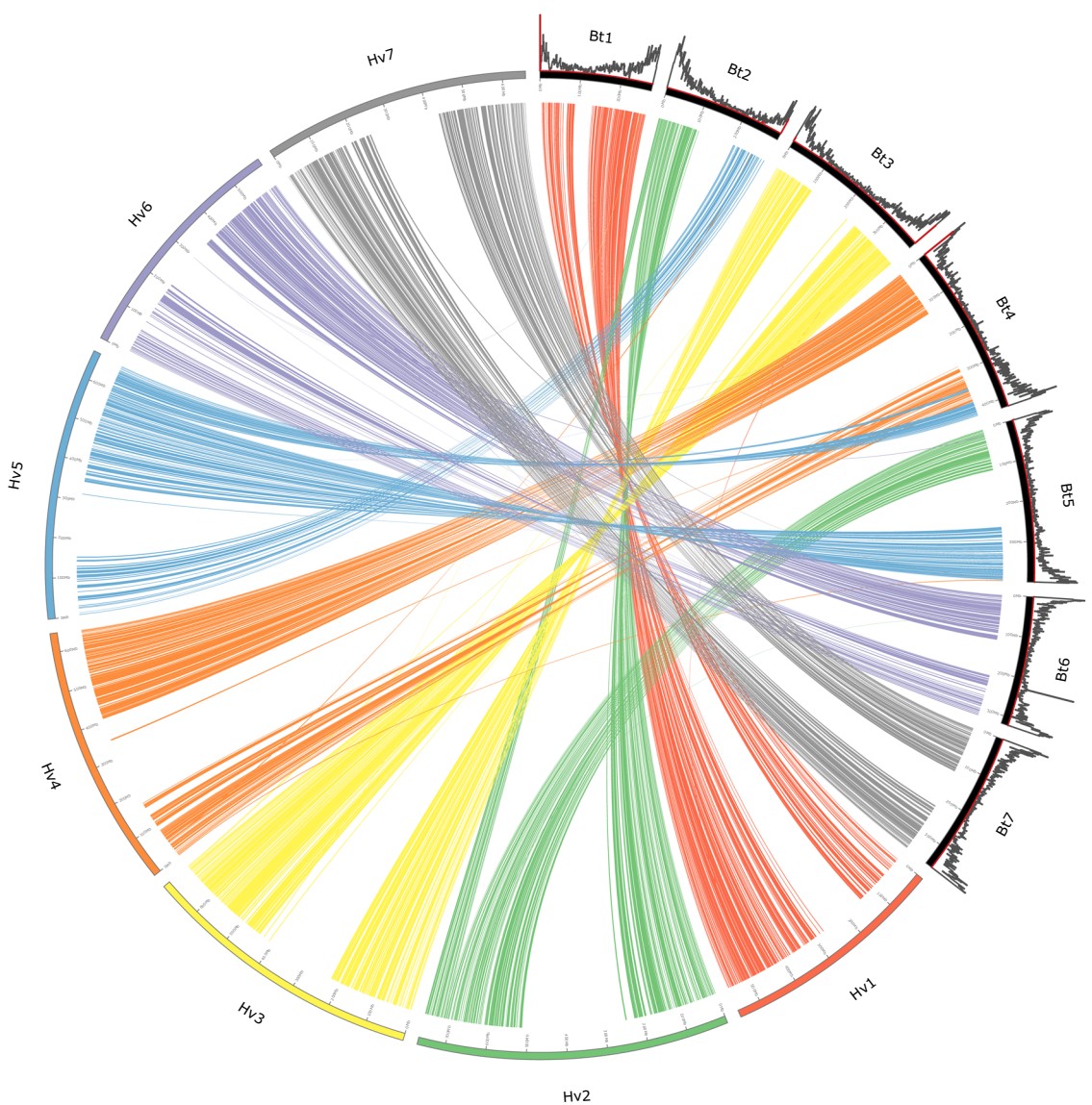

**Fig. 1 Circos synteny plot depicting the synteny between the *Bromus tectorum* draft reference genome and a previously published barley genome.**
Strips and barley chromosomes colored in reference to the corresponding barley chromosomes. Histograms above *B. tectorum* chromosomes are coded black for gene density and red for telomere repeats for each 1 Mb window of the chromosome.

on 121 genotypes to find QTL for PH, VPN, days to first visible joint (J1), FRS, days to 50% ripe seed ($AWN_{50}$) and tiller number using BLINK with three principal components and significance threshold of 0.05 after multiple testing correction. Nineteen QTLs were significantly ($q < 0.05$) associated with PH and reproductive phenology, with one QTL for PH (1), nine QTL for VPN (9), three QTL for J1 (3), nine QTL for FRS (9), and three QTL for $AWN_{50}$ (3). A QTL on *Bt6* (Bt6:8628087) was significant ($q < 0.05$) for all the reproductive phenology related traits except J1, where a second significant ($q = 2.070E-07$) QTL for J1 was at a nearby position (Bt6:8417488) on *Bt6* (Table 2). The QTL on *Bt1* (Bt1:276755111) was significant ($q < 0.05$) for J1, FRS, and $AWN_{50}$ (Table 2). QTLs on *Bt2* (Bt2:9403921) and *Bt7* (Bt7:347158582) were significantly ($q < 0.05$) associated with VPN and FRS (Table 2).

**Plant height**. We identified a QTL that was significantly ($q = 1.360E-05$, Table 2) associated with PH on Bt6:301800092. The QTL explained 16.9% (Table 2) of the phenotypic variation. The MAF at the PH QTL on *Bt6* was 0.44, indicating both allelic

states are common in our panel. Searching the area flanking the QTL associated with PH at Bt6:301800092 on both sides by up to 500 Kb revealed a homolog of *Xanthine Dehydrogenase* (*XDH*) 29 Kb from the QTL and a homolog of *Indole-3-pyruvate monooxygenase YUCCA6* (*YUC6*) 242 Kb from the QTL (Table 3). *XDH* and *YUC6* are promising candidate genes for the QTL associated with PH we identified on *Bt6* as they both are well document to be associated with changes in PH, senescence, and response to drought[28,29]. In rice (*Oryza sativa* L.), overexpression of the *XDH* homolog led to increased PH while under-expression of the *XDH* homolog resulted in reduced PH[28], indicating that homologs of *XDH* would be fitting candidate genes for PH. Indeed, in rice, a GWAS identified *XDH* as a candidate gene for coleoptile length in response to flooding[30] indicating *XDH* homologs may be involved with stem elongation in grasses and thus PH. Furthermore, in Arabidopsis knock-out mutations of the *XDH* gene led to reduced PH[31]. Knock-out mutations of the *YUC6* gene in Arabidopsis was found to increase auxin production in the shoots leading to reduced PH[32]. Further investigation, including gene expression experiments and/or knockout of the

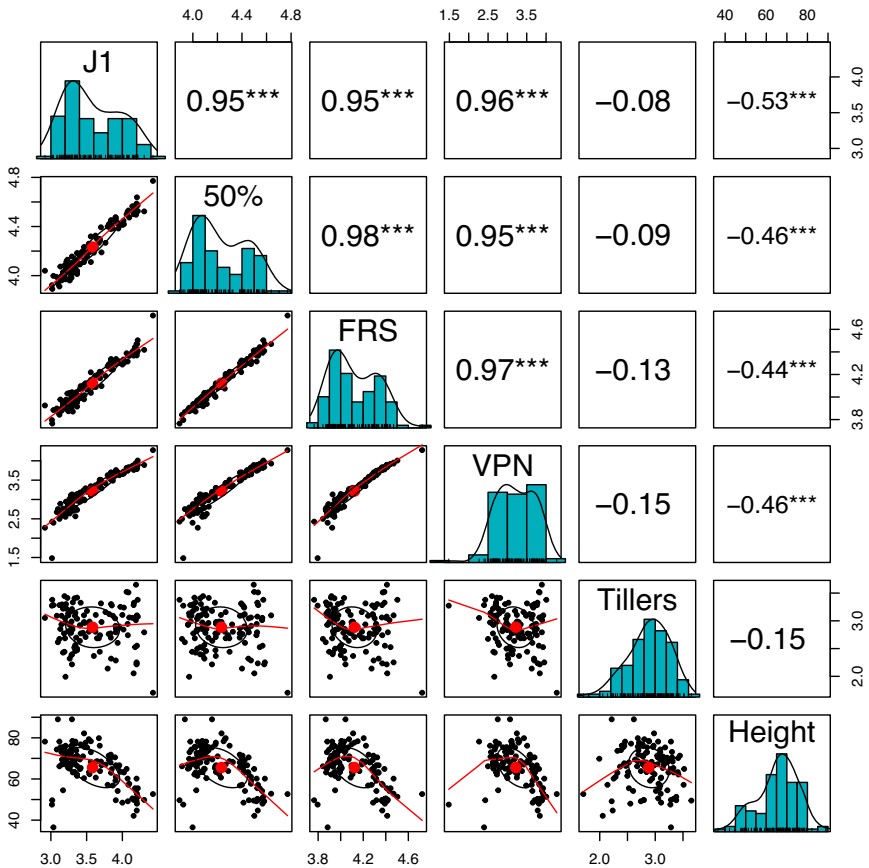

**Fig. 2 Distributions, Spearman correlations, and scatter plots of the genotype BLUEs for height (cm), number of tillers, days until first panicle (VPN), days until 50% rip seed (50%), days until first joint (J1), days until first ripe seed (FRS) used in the GWAS analysis.** Histograms with density ticks and a smoothing line are on the diagonals. The upper diagonal contains the Spearman correlation between BLUEs of traits with "***" denoting a significant level ($p < 0.0001$) of correlation between traits, where $N = 121$ genotypes. The lower diagonal contains scatter plots between traits with center dot, a centroid with a standard deviation of 1 and locally estimated scatterplot smoothing (LOESS) smoothing curve.

**Table 1 Description of the variation and consistency of phenotypic data across lines in greenhouse trials.**

| Trait | H² | BLUE$_\mu$ | BLUE$_{sd}$ | BLUE$_{min}$ | BLUE$_{max}$ |
|---|---|---|---|---|---|
| Height | 0.96 | 65.57 | 9.82 | 36.5 | 89 |
| Tillers | 0.94 | 2.88 (19.05) | 0.37 | 1.70 (5.5) | 3.65 (38.5) |
| VPN[a] | 0.99 | 3.22 (27.89) | 0.49 | 1.48 (4.41) | 4.28 (72.00) |
| J1[b] | 0.98 | 3.58 (38.70) | 0.38 | 0.38 (18.50) | 4.43 (83.77) |
| FRS[c] | 0.96 | 4.12 (62.94) | 0.2 | 3.76 (43.15) | 4.72 (112.5) |
| AWN$_{50}$[d] | 0.97 | 4.23 (70.46) | 0.21 | 3.89 (70.46) | 4.77 (118.1) |

Mean (BLUE$_\mu$), standard deviation (BLUE$_{sd}$), minimum (BLUE$_{min}$), and maximum (BLUE$_{max}$) of best linear unbiased estimates (BLUE) and reliability (H²) of the phenotypic traits measured in the greenhouse.
[a]Days to first visible panicle.
[b]Days to first visible joint.
[c]Days to first ripe seed.
[d]Days to 50% ripe seed.

*XDH* gene homolog in *B. tectorum*, is needed to validate and further understand the underlying genetics controlling the large effect QTL for PH.

Unexpectedly, PH and reproductive phenology timing were negatively correlated, contradicting previous findings that PH and reproductive phenology were positively correlated (i.e., earlier flowering plants do not have as much time to grow)[33–35]. *Lolium perenne* (L.) was also found to have a negative genetic correlation between PH and flowering time[36]. The negative correlation was thought to be the result of selection imposed by grazing, where biological fit plants remained short until they were ready to flower at which point they elongated and flowered quickly[37]. *Bromus*

*tectorum* may also be under similar grazing selection pressure[38]. Competition with crops could also select for taller, fast-growing phenotypes to facilitate competition for space.

Although only a single QTL was uncovered by the GWAS for PH in *B. tectorum*, there is likely many QTL of small effect that GWAS model did not have the statistical power to detect. When GWASs were used to uncover variants associated with complex diseases in humans, the QTLs identified only explained a fraction of the variation compared to what was expected based on heritability estimates[39], becoming known as "the missing heritability problem". Because the missing heritability problem is caused by a lack of power to detect causal variants and by

**Table 2 Marker-trait associations detected by BLINK for plant height (PH) first ripe seed (FRS), first visible panicle (VPN), or 50% ripe seed (AWN$_{50}$); the marker the trait appears to be associated with; the minor allele frequency (MAF), the percent of variation explained by the QTL, and the P-value of the SNP after FDR correction for multiple testing.**

| Trait | Genetic marker | Minor allele frequency | Variation explained (%) | P-value |
|---|---|---|---|---|
| PH[a] | Bt6:301800092 | 0.44 | 16.9 | 1.360E−05 |
| FRS[b] | Bt7:347158582 | 0.29 | 7.2 | 5.280E−09 |
| FRS | Bt1:173453655 | 0.16 | 1.4 | 1.582E−04 |
| FRS | Bt1: 39144935 | 0.22 | 1.8 | 1.786E−04 |
| FRS | Bt3:48964562 | 0.21 | 1.2 | 1.786E−04 |
| FRS | Bt1:276755111 | 0.2 | 4.1 | 2.823E−04 |
| FRS | Bt2:9403921 | 0.25 | 1.9 | 4.166E−04 |
| FRS | Bt4:382201414 | 0.058 | 0.6 | 1.026E−03 |
| FRS | Bt3:4974562 | 0.15 | 4 | 1.836E−03 |
| FRS | Bt6:8628087 | 0.2 | 2.3 | 4.110E−02 |
| VPN[c] | Bt7:70795764 | 0.26 | 16.3 | 3.630E−07 |
| VPN | Bt3:2025814 | 0.31 | 4.3 | 8.210E−07 |
| VPN | Bt4:382636922 | 0.42 | 4.3 | 1.920E−05 |
| VPN | Bt6:8628087 | 0.2 | 2.9 | 4.830E−03 |
| VPN | Bt7:347158582 | 0.29 | 4.7 | 1.046E−02 |
| VPN | Bt2:9403921 | 0.25 | 3.3 | 3.485E−02 |
| VPN | Bt3:324738958 | 0.45 | 3.8 | 3.797E−02 |
| VPN | Bt3:324739133 | 0.49 | 3.4 | 3.797E−02 |
| VPN | Bt3:383345273 | 0.49 | 3.4 | 3.797E−02 |
| AWN$_{50}$[d] | Bt1:276755111 | 0.2 | 6.5 | 6.760E−05 |
| AWN$_{50}$ | Bt5:348443767 | 0.37 | 5.6 | 1.637E−04 |
| AWN$_{50}$ | Bt6:8628087 | 0.2 | 4.8 | 5.973E−03 |
| J1[e] | Bt6:8417488 | 0.35 | 6.7 | 2.070E−07 |
| J1 | Bt1:276755111 | 0.2 | 8.1 | 3.870E−05 |
| J1 | Bt1:169814345 | 0.35 | 4.2 | 1.909E−03 |

[a]Plant height.
[b]Days to first ripe seed.
[c]Days to first visible panicle.
[d]Days to 50% ripe seed.
[e]Days to first visible joint.

inflated measures of heritability, increasing the sample size can partially resolve the heritability problem by increasing the power to detect causal variants. In Atlantic salmon (*Salmo salar* L.), an initial GWAS for age at maturity with a sample size of 1518 only detected a single QTL that explained 39% of the variation in age at maturity[40]. However, a subsequent GWAS study for age at maturity in *S. salar* with a sample size 11,166 uncovered the same QTL previously identified along with 115 other QTLs with smaller effect sizes and low MAFs[41]. If subsequent GWASs are performed for PH in *B. tectorum*, then substantially increasing the number of genotypes in the GWAS would likely uncover smaller effect loci contributing to PH.

**Phenology traits**. Flowering time is an important adaptive trait for ensuring the survival of plants in a broad range of climates, often driving local adaptation[42–44]. In predominantly self-fertilizing species, large effect QTLs control the variation in flowering time[45,46], in contrast to maize where flowering time is controlled by many QTL of small effect[47]. Here, we phenotyped four traits (VPN, J1, FRS, AWN$_{50}$) associated with flowering time to facilitate a GWAS to identify candidate reproductive phenology genes. Days to first visible panicle (VPN) was the first reproductive phenology stage observed and had the highest heritability at 0.99 (Table 1). The largest effect estimated for a significant ($q = 3.630E−07$) QTL for VPN was found at Bt7:70795764, explaining 16.3% of the variation in VPN with a MAF of 0.26. The eight-remaining significant ($q < 0.05$) QTL on *Bt3* (Bt3:2025814), *Bt4* (Bt4:382636922), *Bt6* (Bt6:8628087), *Bt7* (Bt7:347158582), *Bt2* (Bt2:9403921), *Bt3* (Bt3:383345273), *Bt3* (Bt3:324738958) and *Bt3* (Bt3:324739133) explained 4.3, 4.3, 2.9, 4.7, 3.3, 3.8, 3.4 and 3.4% of the phenotypic

variation, respectively, with MAF ranging from 0.2 to 0.49 (Table 2). Small to moderate effect QTL were also underlying the emergence of panicles from the flag leaf in barley[48] with the PVE of QTLs ranging from 1 to 13%.

Developmentally, days to first joint (J1) was the second reproductive phenology trait to occur. Three significant ($q < 0.05$) quantitative trait loci (QTL) were associated with J1. The QTL at *Bt6* (Bt6:8417488; $q = 2.070E−07$) was 12 Kb from a homolog of *Heading Date Repression 1* (*HDR1*) and explained 6.7% of the phenotypic variation with an MAF of 0.35 (Tables 2, 3). The QTL at *Bt1* (Bt1:276755111; $q = 3.870E−05$) explained 8.1% of the phenotypic variation with a MAF of 0.2. The other QTL on *Bt1* (Bt1:169814345; $q = 1.909E−03$) explained 4.2% of the phenotypic variation and had an MAF of 0.35. Although no previous studies have directly mapped genes for time of the visible first joint in a grass species, the high correlations between reproductive phenology traits (Fig. 2) indicate that the genetic architecture of J1 in *B. tectorum* should be like the genetic architecture of other reproductive phenology traits. A QTL mapping study in *Panicum hallii* (Vasey), a perennial primarily self-fertilizing grass species native to the Southwestern United States, revealed only two QTL controlling flowering time with PVEs of 6.4 and 7.4[49], indicating that moderate effect QTL like those found underlying J1 may be underlying reproductive phenology traits in wild populations of self-fertilizing grasses.

The next developmental stage was first ripe seed (FRS). Days to first ripe seed was associated with nine significantly ($q < 0.05$) associated QTL. The leading QTL located on *Bt7* (Bt7:347158582; $q = 1.360E−05$) explained 7.2% of the phenotypic variation of FRS and was common (MAF = 0.29) (Table 2). The remaining QTLs on *Bt7* (Bt7:347158582; $q = 5.280E−09$), *Bt1* (Bt1:173453

**Table 3 Candidate genes within 250 kbp of SNP identified by GWAS, with two ortholog species *Arabidopsis thaliana* and *Oryza sativa*.**

| Trait(s) | Gene | Ortholog Species | Associated SNP[a] | Distance From SNP (Kb) |
|---|---|---|---|---|
| PH[b] | XDH | *O. sativa* | Bt6_301800092 | 29 |
| PH | YUC6 | *A. thaliana* | Bt6_301800092 | 242 |
| FRS[c], VPN[d] | FHY3 | *A. thaliana* | Bt7_347158582 | 149 |
| VPN | FRS7 | *A. thaliana* | Bt3_383280959 | 64 |
| FRS, VPN | FRS6 | *A. thaliana* | Bt7_347158582 | 147 |
| FRS | FRS5 | *A. thaliana* | Bt1_173453655 | 78 |
| FRS | DREB1F | *O. sativa* | Bt1_173453655 | 236 |
| FRS | CKX2 | *O. sativa* | Bt3_48964562 | 55 |
| FRS, VPN | CUL3A | *A. thaliana* | Bt2_9403921 | 29 |
| FRS, VPN | ABCB19 | *A. thaliana* | Bt2_9403921 | 13 |
| FRS, VPN | GA20OX2 | *O. sativa* | Bt2_9403921 | 133 |
| FRS, VPN, AWN50[e] | HDR1 | *O. sativa* | Bt6_8628087 | 212 |
| J1[f] | HDR1 | *O. sativa* | Bt6_8417488 | 12 |
| VPN | PHL | *A. thaliana* | Bt3_2025814 | 240 |
| VPN | BPM1 | *A. thaliana* | Bt3_2025814 | 120 |
| VPN | BPM2 | *A. thaliana* | Bt3_2025814 | 177 |
| VPN | UVR8 | *A. thaliana* | Bt3_324738958 | 113 |
| VPN | UFC | *A. thaliana* | Bt3_324738958 | 64 |
| VPN | VIP2 | *A. thaliana* | Bt3_324738958 | 189 |

[a]Single nucleotide polymorphism.
[b]Plant height.
[c]Days to first ripe seed.
[d]Days to first visible panicle.
[e]Days to 50% ripe seed.
[f]Days to first visible joint.

655; $q = 1.582E-04$), *Bt1*(Bt1:39144935; $q = 1.786E-04$), *Bt3* (Bt3:48964562; $q = 1.786E-04$), *Bt1* (Bt1:276755111; $q = 2.823E-04$), *Bt2* (Bt2:9403921; $q = 4.166E-04$), *Bt4* (Bt4:382201414; $q = 1.026E-03$), *Bt3* (Bt3:4974562; $q = 1.836E-03$) and *Bt6* (Bt6:8628087; $q = 4.110E-02$) explained 1.4%, 1.8%, 1.2%, 4.1%, 1.9%, 0.6%, 4.0% and 2.3% of the variation, respectively (Table 2). The MAF of QTL associated with FRS were lower than those identified for the other reproductive phenology traits, varying from 0.058 to 0.29.

In *Erythranthe laciniata* (A. Gray), a self-fertilizing dicot plant species, populations along an altitudinal gradient differed in their times to FRS with populations from low elevations having the earliest FRS dates and high-elevation populations having the latest FRS dates[50]. The differences in the FRS phenotype between populations of *E. lacinata* are likely driven by moderate to large effect QTL controlling other reproductive phenology trait with PVEs ranging from 9 to 39%[51], exceeding the effect sizes identified for FRS in *B. tectorum* (Table 2). However, the QTL mapping study in *E. laciniata* included both genotypes that require and do not require vernalization[51]. The GWAS study presented here only included *B. tectorum* genotypes that required vernalization, thus major QTL associated with vernalization would not be present. The lack of variation in vernalization genes in the *B. tectorum* genotypes used for the GWAS could explain the absence of large effect QTL underlying FRS in *B. tectorum*.

The final developmental reproductive phenology trait to occur was AWN50, resulting in three significant associations ($q < 0.05$). The leading QTL on *Bt1* (Bt1:276755111; $q = 6.760E-05$) explained 6.5% of the phenotypic variation and had a MAF of 0.2 (Table 2). The other two QTL were located on *Bt5* (Bt5:348443767; $q = 1.637E-04$) and *Bt6* (Bt6:8628087; $q = 5.973E-03$) explaining 5.6% and 4.8% of the phenotypic

variation with MAFs of 0.37 and 0.2, respectively. Although no previous studies have mapped genes for AWN50, the high correlation with other reproductive phenology traits likely means that the genetic architecture underlying the traits will be similar.

Although phenotypic Spearman correlations between reproductive phenology traits are all above 0.95 (Fig. 2), and there is overlap in the QTLs detected (Table 2), the number of associated QTL detected varied from three to nine for reproductive phenology traits (Table 2). The most likely explanation is that slight differences in the phenotype are great enough to change the p-values for association of QTL with a trait. In maize, a study revealed nine QTLs for growing degree days (GDD) to silk while one of the QTL identified for GDD to silk was the only QTL identified by the GWAS for GDD to tassel[52], indicating that it is possible for correlated reproductive phenology traits to yield different number of QTL. Furthermore, the GWAS study presented here uses the GWAS method BLINK, a multi-locus iterative GWAS model[53]. The process of BLINK iteratively updating the model and recalculating p-values means that if the traits are differing, even by a small amount, that different sets of QTLs could be selected early on to build the statistical model used for significance testing thus leading to substantially different sets of QTL being identified for the two correlated traits.

**Survey of candidate genes associated with phenology.** Seventeen genes were identified as punitive candidates for reproductive phenology traits, based on their proximity to QTL (within a 500 Kb interval, spanning 250 Kb on either side of the SNP) and previous functional characterization related to specific phenologies. The most promising of these genes is a homolog of *HDR1*. The proximity of a homolog of HDR1 with QTL from all the maturity-related trait analyzed suggests that *HDR1* influences maturation in *B. tectorum*. In *O. sativa*, *HDR1* promotes *Heading date 1* (*Hd1*) and represses *Early heading date 1* (*Ehd1*) delaying flowering[54]. Knockout mutations or RNA interference of *HDR1* resulted in rice plants that flowered 30 days earlier in long day light conditions, making *HDR1* a promising candidate gene for reproductive phenology traits[54].

Multiple candidate genes were identified for several reproductive phenology-associated QTL, indicating multiple genes may underlie reproductive phenology in *B. tectorum*. The QTL on *Bt2* (9403921) associated with VPN and FRS was 13, 29 and 133 Kb from homologs of *ABC transporter B family member 19* (*ABCB19*), *Cullin-3A* (*CUL3A*), and *Gibberellin 20 oxidase 2* (*GA20OX2*), respectively. Loss of function mutations in *ABCB19*, *CUL3A*, and *GA20OX2* led to longer flowering times, indicating all three promote advancement of reproductive phenology[55–57]. Coincidentally, a gene near another QTL associated with FRS is known to interact with *GA20OX2*. The QTL on *Bt1* (173453655) associated with FRS was 78Kb from orthologs of *FAR1-RELATED SEQUENCE 5* (*FRS5*), a punitive transcription factor regulating far-red light control of development[58], associated with adaptation to photoperiod[59], and *dehydration-responsive element-binding protein 1 F* (*DREB1F*), a putative transcription factor, when upregulated, represses gibberellic acid (GA) biosynthesis catalyzed by *GA20OX* genes. The upregulation of *DREB1F* causing shorter plants with delayed flowering[60], indicates that *DREB1F* and *GA20OX* can act in an epistatic manner to control reproductive phenology. A likelihood ratio test of the interaction between the FRS QTLs on *Bt2* (9403921) and *Bt1* (173453655) revealed epistasis is likely ($X^2 = 3.71$ df = 1, $p = 0.054$), indicating *DREB1F* and *GA20OX* are interacting to control FRS in an epistatic manner. The epistatic interaction also indicates further that *DREB1F* and *GA20OX* are likely the genes actively

controlling FRS at the QTL on *Bt2* (9403921) and *Bt1* (173453655).

The QTL associated with VPN on *Bt3* (2025814) was 120Kb, 177Kb and 240Kb from homologues of *BTB/POZ and MATH domain-containing protein 1* (*BPM1*), *BTB/POZ and MATH domain-containing protein 2* (*BPM2*) and *PHYTOCHROME-DEPENDENT LATE-FLOWERING* (*PHL*), respectively. *PHL* triggers flowering under long-day conditions by repressing the phytochrome b (*PHYB*) and the constans (*CO*) genes[61]. *BPM1* and *BPM2* are transcription factors of the *BTB/POZ and MATH domain-containing protein* (*BPM*) gene family and are involved in regulating flowering time by making proteins that are part of the Cullin E3 ubiquitin ligase complexes that include *CUL3A*[62]. A likelihood ratio test did not detect epistasis ($X^2 = 0.02$, df = 1, $p = 0.901$) between the QTL on *Bt2* (9403921) near *CUL3A* and the QTL on *Bt3* (2025814) near *BPM1* and *BPM2*.

A QTL on *Bt3* (324738958) associated with *VPN* was located 64Kb, 113Kb and 189Kb from homologs of *UPSTREAM of FLC* (*UFC*), *Ultraviolet-B receptor 8* (*UVR8*) and *Early Flowering 7* (*VIP2*), respectively. *UVR8*, *VIP2* and *UFC* are involved with the regulation of *Flowering Locus C* (*FLC*) in *A. thaliana*[63–65], suggesting one mechanism maintaining genetic variation in *B. tectorum* flowering time is the regulation of FLC expression. Homologs of four genes in the *FHY3/FAR1* gene family were near QTL found associated with maturity traits indicating the *FHY3/FAR1* gene family is part of a mechanism maintaining variation in maturity traits in *B. tectorum*. Homologues of *FHY3* and *FRS6* were found 147 and 149 Kb from a QTL on *Bt7* (Table 3), respectively. A homologue of *FRS7* was near the QTL on *Bt3* associated with VPN (Table 3) *FRS5* discussed earlier is also member of the *FHY3/FAR1* gene family. The *FHY3/FAR1* gene family is comprised of 14 homologous genes, regulating transcription as a response to far red light[58]. *FHY3*, *FRS6*, *FRS7* have been demonstrated to regulate flowering time in *A. thaliana*[58,66,67] making members of the *FHY3/FAR1* good candidate genes for maturity traits.

In addition, a *Cytokinin Dehydrogenase 2* (*CKX2*) homolog was identified as a candidate gene for the QTL at Bt3:48964562 associated with FRS. *CKX2* catalyzes the oxidation of cytokinins[68]. Cytokinins have been shown to regulate flowering time via transcriptional activation of *Twin Sister of FT* (*TST*)[69] and overexpression in members of the *Cytokinin Dehydrogenase* (*CKX*) gene family have shown to delay flowering time in long day conditions[68] indicating that the *CKX2* homolog could be controlling maturity traits through the regulation of cytokinins in *B. tectorum*.

**Implications of the reference genome and phenology GWAS.** The reference quality genome we have assembled is an invaluable resource for understanding the fundamental genetic controls that have facilitate one of the most successful invasive weeds in North America. Understanding the genetic basis of adaptive traits in *B. tectorum* will lead to improved management strategies. Using the genome we explored the genetic underpinnings of reproductive phenology traits in *B. tectorum*, revealing pathways and mechanisms contributing to adaptive plasticity which directly contributes to the species invasive spread across a wide range of environments. Indeed, our GWAS uncovered genetic mechanisms contributing to plasticity within the species. We identified QTL for maturity traits that were near candidate genes responsible for controlling the photoperiod pathway, plant hormone regulation, and transcription factors triggered by far-red or UV light.

Our study indicates that not only is the genome very similar to barley (Fig. 1), but the adaptive control of reproductive phenology also closely mirrors barley. Both domesticated and wild barley are adapted to latitudinal clines, where reproductive phenology is controlled by genes responding to environmental cues[70]. Our identification of a *GA20OX* ortholog as a candidate gene for reproductive phenology corroborates previous work, where *GA20OX* loss of function was found to reduce PH and delay flowering[35,71,72]. Interestingly, *GA20OX* orthologs are semi-dwarfing genes implicated in the green revolution[73]. The semi-dwarfing gene we identified as a candidate gene could also explain the negative correlation between reproductive phenology traits and PH. In addition, wild barley was found to adapt using variation in photoperiod response genes[70], supporting our findings where the *FHY3/FAR1* photoperiod receptor gene family was implicated in controlling reproductive phenology.

Although *GA20OX* is a known controller of PH, its homolog was not identified by the GWAS as associated with PH. Plant height is an adaptive trait with large effect QTL conserved for maintaining phenotypic variation within in and between species of Poaceae[74]. However, GWASs for PH in predominantly self-fertilizing grass species have revealed variable effect sizes of the leading QTLs (QTL with highest PVE for a trait in a GWAS) between species, ranging from 1 to 23%[75–77], demonstrating that the genetic architecture of plant height varies among self-fertilizing grass species. Our study indicates that *B. tectorum* uses at least one moderate to large effect locus to control PH as we only identified a single locus explaining 16.9% of the phenotypic variation. If mutations in *B. tectorum* are slightly deleterious then it would be expected that *B. tectorum* would need to use a small number of large effect QTL to facilitate local adaptation in the presence of divergent stabilizing selection[78]. Thus, the identification of the single moderate to large effect locus for PH may indicate the presence of deleterious mutations and divergent stabilizing selection between locales in *B. tectorum*, although further experiments would be needed to validate the selection and deleterious mutations imposed on *B. tectorum*. The homologs of *XDH* and *YUC6* as candidate genes reflected the involvement of stress response and hormonal regulator genes controlling PH as found in other phylogenetically proximal grasses, such as wheat[28], barley[29], and oats[30].

Our study indicates *B. tectorum* is armed with a complex array of genetic mechanisms to create adaptive variation underlying reproductive phenology and PH which has facilitated its invasion into N. America and suggests that it is likely to continue to spread north into western Canada as climate change facilitates range expansion[79]. Identifying candidate *FHY3*, *FRS5*, *FRS6*, *FRS7*, *ABCB19*, *UVR8*, and *PHL* genes that all act in response to light stimulus indicate that photoreceptor genes are critical for controlling variation in flowering time in *B. tectorum*. Photoreceptors controlling reproductive phenology will result in phenotypic plasticity because light signals of the environment will influence the underlying pathways[80]. Genetic control of traits was quite high, ranging from 0.99 to 0.94 for VPN and tiller number, respectively.

Furthermore, our results indicated the presence of small to moderate effect QTL controlling reproductive phenology traits, rather than singular large effect loci. The high heritability and moderate effect QTL detected for adaptive traits indicate that *B. tectorum* has already been adapting and will continue to adapt to a wide range of environments utilizing moderate-sized QTL—as genetic recombination is very rare. Further investigation into the plasticity and regulation of flowering time in *B. tectorum* is needed to understand how local populations respond to stress or climate variation, and how the genetic variation we have discovered confers success in the arid, dry southern reaches of the American southwest and northern Mexico north to western Canada.

## Methods

**Plant Material and DNA extraction for whole genome sequencing.** For whole genome assembly, a single plant from the *B. tectorum* accession (FMH10) was grown hydroponically in an isolated, disease-free growth chamber under a 12-h photoperiod. Growing temperatures ranged from 18 °C (night) to 20 °C (day). The hydroponic growth solution was based on MaxiBloom® Hydroponics Plant Food (General Hydroponics, Sevastopol, CA, United States) at a concentration of 1.7 g/L. FMH10 is a common clade accession[11]. In preparation for PacBio CLR sequencing, high molecular weight DNA was extracted from 72-h dark-treated leaf samples using a CTAB-Qiagen Genomic-tip protocol as described by Vaillancourt and Buell[81].

**Whole genome sequencing.** For whole-genome sequencing, large-insert SMRTBell libraries (>20 kb), selected using a SageElf (Sage Science, Inc., Beverly, MA, USA), were prepared according to standard manufacture protocols and sequenced at the BYU DNA Sequencing Center (Provo, UT, USA) using P6-C4 chemistry on a Sequel II instrument (Pacific BioSciences, Menlo Park, CA, USA). For whole genome polishing, DNA was sent to for Illumina HiSeq (2 × 150 bp) sequencing from standard 500-bp insert libraries. Trimmomatic v0.35[82] was used to remove adapter sequences and leading and trailing bases with a quality score < 20 or with an average per-base quality of 20 over a four-nucleotide sliding window. After trimming, any reads shorter than 75 nucleotides in length were removed. Raw PacBio and Illumina reads have been deposited in GenBank.

**Genome assembly, polishing, and Hi-C scaffolding.** A primary assembly of *B. tectorum* accession FMH10 was constructed using Canu v1.9[83] with default parameters (corMhapSensitivity = normal and corOutCoverage = 40). The primary assembly was polished twice with Illumina short reads using Arrow from the GenomicConsensus package in the Pacific BioSciences SMRT portal v5.1.0 followed by a single round of insertion/deletion correction using PILON v0.22[84]. The average read depth for the genome assembly was 68.6. Fresh leaf tissue from a single 3-week-old FMH10 plant was sent to Dovetail Genomics LLC (Santa Cruz, CA, USA) in preparation for construction of an Omni-C™ proximity-guided final chromosome-scale assembly. The Omni-C™ technology uses an approach to Hi-C library preparation via DNA digestion with a non-specific endonuclease to increase uniformity and genomic coverage (https://dovetailgenomics.com/omni-c/). The libraries were prepared using a standard Illumina library prep followed by sequencing on an Illumina HiSeq X in rapid run mode. The HiRiSE™ scaffolder and the Omni-C™ library-based read pairs were used to produce a likelihood model for genomic distance between read pairs, which was used to break putative miss-joins and to identify and make prospective joins in primary contig assembly to produce the final chromosome scale reference assembly.

**GWAS plant materials collection.** *Bromus tectorum* genotypes were obtained from Genome Resources Information Network (GRIN), field samples in eastern Washington collected by Jon Witkop and Amber Hauvermale in 2015 as part of the "Regional Approaches to Climate Change for Pacific Northwest Agriculture" a multi-disciplinary project that aims to mitigate the impacts climate change has on agriculture[85], and samples from natural areas in Montana (contributed by Lisa Rew, Montana State University) with 11, 64, and 46 samples, respectively. Each genotype was grown for one generation in the greenhouse to increase seed and verify purity. Six replicates of each line were vernalized at 4 °C with 10 h light per day for 53 d, then planted in 1.4-liter square pots. Supplemental lighting was used to keep day lengths at least 15 h per day.

**Phenotyping.** Plants in the greenhouse were observed daily to record reproductive phenology-associated phenotypes. The phenotypes measured included days until first panicle visible (VPN, Feekes 10.1), days to first joint (J1, Feekes 6), days until first mature seed (FRS), days until 50% of seeds dry with awns angled outward (AWN$_{50}$), number of tillers, and height of the tallest panicle (PH). Number of tillers were counted for each plant at the end of the experiment before harvest. The height of the tallest panicle was measured in centimeters, measuring from the base of the plant to the tip of the longest panicle.

**GWAS DNA extraction and resequencing.** DNA was extracted using a bromide CTAB protocol as previously described[86]. Samples were diluted to a 50 ng/µl concentration. Genotyping by sequencing libraries were prepared for each sample by LGC Genomics (Berlin, Germany) following Elshire et al.[87] using the *Msl1* restriction enzyme. Barcode adapters were ligated to each sample and the samples were put into 48-plex library plates. The polymerase chain reaction was used to amplify samples on the plates which were then sequenced using a single lane of Illumina NextSeq 500 V2. Approximately 1.5 million (2 × 150 bp) reads were generated per sample.

After sequencing, all the library groups were de-multiplexed with bcl2fastq v2.17.1.14 (https://support.illumina.com/sequencing/sequencing_software/bcl2fastq-conversion-software.html) software allowing for up to two mismatches on the barcodes. Library groups were de-multiplexed further into separate samples according to the inline barcodes, where no mismatches were allowed. The adapter barcodes were clipped and reads <20 bases in length were discarded as were any reads where the 5′ end did not match the restriction enzyme cutting motif. Reads were quality trimmed from the 3′ end so that the average Phred quality score across ten neighboring bases >20.

**GWAS SNP calling.** De-multiplexed filtered reads for each sample were aligned to the de novo reference genome using BWA mem[88] with default settings for paired end reads. SAM files generated from the alignments were converted to BAM files and sorted using SAMTOOLS[89]. The mpileup and call functions from Bcftools[90] were used to call SNPs. The vcf file generated from Bcftools was filtered to only SNP variants were kept, minor allele frequency (MAF) > 0.05, Missing Alleles <25% and a QUAL score of at least 30 for each SNP, using Bcftools[89]. Although the vcf file was not explicitly filtered for read depth, read depths were adequate with a minimum read depth of 54 and a median read depth of 808. Scripts in R statistical programming language were used to read the vcf file into an allelic dosage table and filter out markers with more than two alleles or more than 5 heterozygous calls (*B. tectorum* is an autogamous species). Missing calls were imputed with a k$^{th}$ nearest neighbor imputation using the "impute" package[91] from the Bioconductor project[92] in the R statistical programming language[93].

**Linkage-disequilibrium.** Linkage-disequilibrium (LD) was calculated on a pairwise basis between SNP on the same chromosome within 3 Mb using 1500 randomly sampled SNP from each chromosome. LD, measured as $r^2_{sv}$, was calculated using the method described by Mangin et al.[94], that calculates the Pearson correlation between SNP but corrects for population structure and kinship in LD calculations implemented in the "LDcorSV" R package[95]. A genomic kinship matrix was estimated from the SNP data using the method from VanRaden[96] implemented in GAPIT[97]. Landscape and Ecological Analysis (LEA)[98] was used in R to determine the optimal number of ancestral groups, and then calculate the admixture of each of these ancestral groups. Values of K, ranging from 1 to 20, were evaluated using the snmf function in LEA for cross entropy in ten replications and the lowest K near the lowest cross entropy was selected as the optimal K. The kinship matrix and population components from snmf described above were used with LDcorSV to correct for population structure, with distances above 3 Mb being discarded. The data from all the chromosomes was pooled together after filtering SNP by distance. The nlrq function of the quantreg R package[99] was used estimate an asymptotic decay by physical genetic distance for the 90th percent quantile of LD values. The LD decay was defined as the distance required for the LD (90th percent quantile of corrected $r^2$) to drop from its initial starting point to half-way between the starting point and the lower asymptotic limit (LD$_{90,1/2}$). LD$_{90,1/2}$ as a measure was found by simulation to be a more accurate estimate then $r^2 = 0.1$ as a measure of LD decay[100].

**GWAS analysis.** GWAS was performed using the Bayesian-information and Linkage-disequilibrium Iteratively Nested Keyway (BLINK) algorithm[53] with the BLUEs for each phenotype as the response variables and the numeric SNP matrix as the genetic data. BLINK was ran using the GAPIT package[97] in R. Three principal components were chosen to be included as a covariate in the BLINK which represented the smallest number of principal components that controlled inflation on the P-P diagnostic plots generated by GAPIT. Potential candidate genes within a 1 Mb window, based on LD (Fig. S2) for each of the genomic regions identified in the GWAS analysis were manually identified from the previously described annotated gene set with functional annotations indicating association with maturity traits.

**Transcriptome assembly and genome annotation.** RNA-Seq data (2 × 150 bp Illumina reads), derived from a bulk tissue sample consisting of 7-d-old seedlings and leaf, roots, and stems from hydroponically grown *B. tectorum* (FMH10) plants, was trimmed using Trimmomatic[72] and aligned to Omni-C reference assembly using HiSat2 v2.0.4[101] with default parameters and max intron length set to 50,000 bp. The resulting SAM file was sorted and indexed using SAMtools v1.6[89] and assembled into putative transcripts using StringTie v1.3.4[102]. The quality of the assembled transcriptome was assessed relative to completeness using BLAST comparisons to the reference *Brachypodium distachyon* L. (ftp://ftp.ensemblgenomes.org/pub/plants/release-37/fasta/brachypodium_distachyon/pep/).

Prior to annotation with MAKER2 v2.31.10[103], RepeatModeler v1.0.11[104] and RepeatMasker v4.0.7[105] were used to identify repetitive elements in the final reference assembly, relative to RepBase libraries v20181026; www.girinst.org. Transcriptome evidence for the annotation included the de novo transcriptome for *B. tectorum* as well as the cDNA models from *Brachypodium distachyon* (v 1.0; Ensembl genomes). Protein evidence included the uniprot-sprot database (downloaded September 25, 2018) as well as the peptide models from *B. distachyon* (v 1.0; Ensembl genomes). Repeats within the reference assembly were masked based on the species-specific sequences produced by RepeatModeler. For ab initio gene prediction, *B. tectorum*-specific AUGUSTUS gene prediction models were provided to MAKER as well as rice (Oryza sativa L.)-based SNAP models. Benchmarking Universal Single-Copy Orthologs (BUSCO) v3.0.2[106] was used to assess the completeness of the final assembly using the Embryophyta odb10 dataset, with the "–long" argument.

Syntenny analysis was performed using McScanX[27] with results of a pBlast with default settings between the predicted genes of our OMNI-C *B. tectorum* and the barley reference genome. Gene density was calculated for each 1 Mb window of the *B. tectorum* reference genome. Telomere density was also calculated for 1 Mb windows using the Blast results between the nucleotide sequence of the *B. tectorum* reference genome and "TTAGGG" the telomeric repeat of most plant species[107] repeated four times. Custom Python code was used to create simplified GFF files to run McScanX and format the results for creating a circle plot. Circos[108] was used to create a circle plot using the results from the protein blast between barley and *B. tectorum*.

**Statistics and reproducibility**. Three replicates, or blocks, were placed in one of two greenhouses with the same $N = 121$ genotypes in each. The plants were arranged in a completely randomized block design, with each block grown on a separate greenhouse bench. linear mixed effect (LME) and generalized linear mixed effect (GLME) models were fit using the "lme4" R package[109] using lmer and glmer functions, respectively. LMEs were used for traits that are continuous measurements, GLMEs with a sqrt link function were used for count data. The linear model for the LME is:

$$y_{ijk} = \mu + g_i + \beta(E)_{jk} + E_k + (gE)_{ik} + \epsilon_{ijk} \tag{1}$$

Genotype $\sim N(0, \sigma^2_g)$
Block within Greenhouse $\sim N(0, \sigma^2_{\beta(E)})$
Greenhouse $\sim N(0, \sigma^2_E)$
Genotype × Greenhouse $\sim N(0, \sigma^2_{(gE)})$
Error $\sim N(0, \sigma^2_e)$

Where $y_{ijk}$ the phenotype, $\mu$ is the mean, $g_i$ is a random effect due to genotype, $\beta(E)_{jk}$ is a random effect of the block within greenhouse, $E_k$ is the random effect of greenhouse, and $\epsilon_{ijk}$ is the error term. Broad Sense Heritability for each trait was estimated according to Cullis et al.[110].

$$H^2 = 1 - \frac{\overline{v}Blup}{2\sigma^2_g} \tag{2}$$

where $\overline{v}Blup$ is the mean variance of a difference of two blups and $\sigma^2_g$ is the variation of the random effect for genotype. For count data where a generalized mixed linear model was used heritability was calculated in the latent (transformed) distribution. BLUEs were calculated for the GWAS using the model for heritability modified to have genotype set as a fixed effect instead of a random effect, and a log link function instead of square root link function.

The statistical testing for the GWAS analyses and ad-hoc analyses for the GWAS all used BLUEs generated from [1] with $N = 121$ genotypes. Interactions were tested for when protein products of two genes near two QTL are known to interact through previous studies are found near significant ($q < 0.05$) QTL for the same trait. The following models for each trait where QTL associated with the trait were fit using the lme4qtl[111] R package. For kinship matrix required we used the "Van-Raden" method implemented in GAPIT R package on our full SNP matrix to generate a kinship matrix. Models were solved using restricted maximum likelihood (REML) for calculating percent variation explained by SNPs while maximum likelihood (ML) was used to solve the models when the models were used for likelihood ratio test of an interaction[112].

$$\mathbf{y} = X\mathbf{b} + Z\mathbf{u} + M\mathbf{v} + \mathbf{e} \tag{3}$$

$$\mathbf{y} = X\mathbf{b} + Z\mathbf{u} + M\mathbf{v} + (\mathbf{m}_i \odot \mathbf{m}_j)v_{ij} + \mathbf{e} \tag{4}$$

where $\mathbf{y}_{nx1}$ is a vector of the standardized BLUEs, $X_{nx3}$ is a matrix of fixed effects which in our case is the first three principal components of the full SNP matrix with the corresponding vector of coefficients $\mathbf{b}_{3x1}$. $M_{nxp}$ is a numeric SNP matrix comprising of the significant ($q < 0.05$) SNP found by BLINK in the GWAS, $\mathbf{v}_{pxn}$ is the vector of corresponding fixed-effect coefficients for the SNP found by Blink. $(\mathbf{m}_i \odot \mathbf{m}_j)$ is a nx1 vector is the product of component wise vector multiplication of the SNPs being tested for an interaction between $SNP_i$ and $SNP_j$ with $v_{ij}$ a scalar as its fixed effect coefficient. $Z_{nxn}$ is an incidence matrix representing the genetic relationship between genotypes and $\mathbf{u}_{nx1}$ is a vector of random polygenic effects that follows the distribution $N(0, \sigma^2_g A_{nxn})$ where $A_{nxn}$ is the additive relationship matrix. $\mathbf{e}_{nx1}$ is a vector of residual error that follows the distribution $N(0, \sigma^2_e I_{nxn})$.

The likelihood ratio tests were implemented using the *anova* function in r on the models fit for [3] and [4]. Because only one interaction was tested at a time the [3] was nested in the more general model [4]. The *anova* function was used with both fit models are the input. The *anova* function calculates the log-likelihood of each model to calculate the likelihood ratio that is tested using a chi-sq test with one degree of freedom ([4] has one more parameter then [3]). Significant results ($p < 0.05$) indicate the more general model (model with SNP interaction [2]) is more likely to be explained by the data thus there is likely an interaction between those SNP.

Percent variation explained by QTL was calculated for each QTL identified by BLINK for each trait using the following formula for each QTL in model [1]. For

each trait the markers were fit to a model simultaneously to estimate coefficients.

$$h^2_{qtl} = \frac{\sigma^2_{qtl}}{\sigma^2_y} = \frac{(\hat{B}^*_{qtl})^2}{\sigma^2_y} = \frac{Var(\hat{B}P)}{\sigma^2_y} = \frac{Var(P)\hat{B}^2}{\sigma^2_y} = \frac{2f(1-f)\hat{B}^2}{\sigma^2_y} \tag{5}$$

where $h^2_{qtl}$ is the percent of phenotypic variation explained by a given QTL, $\sigma^2_y$ is total phenotypic variation, $\sigma^2_{qtl}$ is the phenotypic variation explained by the QTL, $\hat{B}$ is the estimated effect of a marker ($v_i$) from equation [3], P is the allele frequency which follows a binomial distribution with 2 trials and probability p, $f$ is the minor allele frequency (1-p). By standardizing the BLUEs, $\sigma^2_y$ is set to 1 thus simplifying the equation to:

$$h^2_{qtl} = 2f(1-f)\hat{B}^2 \tag{6}$$

**Reporting summary**. Further information on research design is available in the Nature Research Reporting Summary linked to this article.

## Data availability
The raw sequences used for the *B. tectorum* genome assembly are deposited in the National Center for Biotechnology Information (NCBI) Sequence Read Archive database under the BioProject PRJNA728981 with the following accession numbers: SRR14498212–SRR14498217 (PacBio reads), SRR14578284–SRR14578290 (Hi-C reads), SRR14578282 (Transcriptome) and SRR14498209–SRR14498211 (Polishing short reads). The raw reads for the resequencing panel are found in BioProject PRJNA728981 with the following NCBI accession numbers: SRR15308470–SRR15308851 (resequencing panel). Genome browsing and bulk data downloads, including annotations and BLAST analysis of the final proximity-guided assemblies are available at CoGe (https://genomevolution.org/coge/) with genome ID: id64356. Source data pertaining to general information on the genotypes is available in Supplementary Data 1 and source data of the BLUEs used to produce Fig. 2 and perform GWAS analyses is available in Supplementary Data 2. Genotype files (VCF, numeric SNP matrix), GWAS summary statistics are all freely available for download in a figshare repository associated with this manuscript. (https://doi.org/10.6084/m9.figshare.c.6419786.v1).

## Materials availability
All the genotypes that were phenotyped and genotyped in this manuscript are available upon request.

## Code availability
The custom scripts in R and Python for the phenotypic analysis, automating bioinformatics, and performing GWAS analyses are freely available without restriction on the figshare repository associated with this manuscript (https://doi.org/10.6084/m9.figshare.c.6419786.v1).

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

## Acknowledgements
We would like to acknowledge Amber Hauvermale for providing sequence data on *B. tectorum* that was used to conceive the experiment but was not included in analyses present in the manuscript. We would also like acknowledge Lisa Rew for providing seed for the *B. tectorum* genotypes from Montana. This work was funded in part by the Washington Grain Commission and by U.S. Department of Agriculture National Institute of Food and Agriculture grant No. 2017-68002-26819, the Washington Grain Commission R. J. Cook Chair Endowment, and the USDA National Institute of Food and Agriculture, Hatch project 1017286.

## Author contributions
S.R.R. wrote the manuscript, designed experiments and analyzed data. P.J.M. assembled and annotated the reference genome, analyzed data and edited the manuscript. C.E.C. contributed to the assembly and annotation of the reference genome and edited the manuscript. I.C.B. contributed to writing the manuscript, designed experiments and edited the manuscript.

## Competing interests
The authors declare no competing interests.

## Ethics approval and consent to participate
All relevant permissions were obtained for the collection of *B. tectorum* genotypes. The methods carried out in our manuscript were in accordance with the local, national, and international guidelines and regulations.
