## [Peer Review File · Communications Biology]

Reviewers' comments:

Reviewer #1 (Remarks to the Author):

From the perspective of invasive weeds, this manuscript first assembled a high-quality chromosome-scale reference genome and found a highly syntenic relationship with barley genome. Using GWAS, the authors identified nineteen QTLs and seventeen candidate genes related to reproductive phenology, which was important for understanding the genetic mechanisms underlying the plasticity of invasive weeds. Generally, this is a nice work.

The main problem in this manuscript is that RESULT only listed the analysis results, lacking the support of experimental data. Especially in the "Phenology traits" section, the authors should explain the important conclusions rather than only list all the contents in table 2. Moreover, since there was more than one QTL per chromosome, an additional column of positional information (such as Bt6:8628087) should be added in table 2 to resolve the ambiguity of lines 114-120.

In addition, there were still some problems need to be confirmed in this manuscript.

1> In lines 94-95, "of" should be added between "correlation" and "Best" in the sentence "Broad Sense Heritability (Reliability), the mean, standard deviation, min, max and correlation Best Linear Unbiased Estimates (BLUES) were calculated to ...".

2> In line 106, the number "-0.51" differed from "-0.53" labeled in Figure 2.

3> In lines 342-343, missing close parenthesis.

4> In figure 1, I was not sure if changing the order of the chromosomes is better to show the results. I thought the authors should add a ruler or specification related to the orientation of the chromosomes.

5> In table 2, the sentence "where trait includes plant height (PH) (FRS), (VPN), or (AWN50)" was incorrect.

6> All the figures should be changed to a vector graph to increase clarity.

Reviewer #2 (Remarks to the Author):

This manuscript presented WGS and GWAS works for *B. tectorum*. The results are interesting and I recommend to publish this paper as it would provide a pool of valuable genomic data for the people in the weed science to use for subsequent genomic and genetic studies.

Few specific comments:

L127: comma after "In rice (*Oryza sativa* L.)"

L139. Negative correlation between PH and phenology traits was also report by (<https://link.springer.com/article/10.1007/s10681-020-02686-8>)

L163: First time use this acronym so should spell it out here.

L154-157: do not need to include P values as you put them in the Table already.

L185: while 500 kb was chosen as the interval for searching for candidate genes, in Table 3 it states that the candidate genes are within 250 kb?

L187: needs a dot to finish the previous sentence.

L195-204: FRS in the paper cited (no.49) is the Far-red response mutant. Do you mistaken it with the RFS acronym? If not, why you mentioned only three genes as potential candidate first and then later mentioned something else could be candidate? Did you blast the sequence to know if FRS5 is near the Bt2?

Is it what you meant that for RFS QTL on Bt1, there are two candidates gene including RFS5 and DREB1F, but DREB1F is the more likely candidate?

L227: change the citation format here

L239: in=is?

L254: you cited an oat paper, but can add barley papers as this gene is important in barley as well, and there are tens of paper about that in barley

Overall questions: why PH had a wide variation and a high heritability as well but only 1 QTL detected? It is hard to believe that only 1 QTL control height in *B. tectorum*. Is there any other papers

in literature map and identify only 1 QTL for height in weeds? I meant is it common in weeds to have QTL controlling height? Barley has >10 and these two are similar so is there any way to explain this? The correlation between VPN to FRS and Awn50 was very similar (0.97 to 0.98), however, the number QTL detected for VPN and FRS are 3X more than Awn50. Do you have any explanation? What is the coverage of the sequencing works (is it 40X)?

Reviewer #3 (Remarks to the Author):

Revolinski et al. present a study exploring the *Bromus tectorum* genome and potential genes associated with plasticity in traits that have been recognized as important to the species success as an invasive. The authors present a high-quality genome that will serve as a valuable resource to multiple communities of scientists, especially weed genomicists. The remainder of the study is promising in the highlighting of potential genes that may contribute to traits identified as instrumental in the success of cheatgrass as an invasive.

Overall, I found much of the paper to be straightforward. The introduction seems to be lacking, but that may be partially an artifact of the format of this journal with shorter articles. Below I mention some additional information that would be helpful in the introduction.

There were some portions of the Results that read like Discussion (Lines 129-140 and 188-192). Again, this may be a format of the journal issue. Other articles in the journal do combine these two sections.

It was unclear what coverage cutoff was used in the GWAS to identify SNPs (Lines 359-362). This would be valuable information to have in the paper.

Figure 3 seems like a supplemental figure. There is not a lot the reader gets from looking at the figure that would not be more easily understood from text.

I was surprised that flowers/inflorescences were not used in the annotation given how important phenology is to much of the interpretation of the study. There is no report in the main text or the supplement regarding number of genes annotated or additional annotations for other features (like transposons). It would be valuable to have this included in a table somewhere.

Line 12: Need a "the" between "outcompete" and "native".

Line 19: No ", " needed after "species".

Lines 39-45: It was stated that cheatgrass likely originated in the US through a bottleneck, but this section is discussing genetic diversity of ancestral populations. Are these the original founding populations or farther back in time? Is there evidence of other instances of gene flow from native populations since the original introduction? Some more description and clarity of the known history of invasion and genetic diversity of invasive and native populations would be helpful here.

Lines 45-48: This sentence is a bit unclear. Plasticity is the variable response to a range of factors and not necessarily adaptive. Unless this is meant to say that exhibiting plasticity itself is adaptive here. I think this is an issue of the sentence structure tripping me up.

Line 57: North American

Line 58: Should be Poaceae.

Line 63: Triticeae is a tribe, not subfamily.

Line 202: GA needs to be written out.

Line 265: This seems like an overstatement. This was one study for one species. This is overgeneralizing what selfing species may do. There are other selfing grasses with phenotype and genomic data sets you compare to see if this is a trend or not.

Line 277: Need a period after reference.

Line 304: Reference issue.

Referee expertise:

Referee #1: GWAS, plant genome sequencing

Referee #2: GWAS, BLUEs

Referee #3: invasive species genomics

Reviewers' comments:

Reviewer #1 (Remarks to the Author):

From the perspective of invasive weeds, this manuscript first assembled a high-quality chromosome-scale reference genome and found a highly syntenic relationship with barley genome. Using GWAS, the authors identified nineteen QTLs and seventeen candidate genes related to reproductive phenology, which was important for understanding the genetic mechanisms underlying the plasticity of invasive weeds. Generally, this is a nice work.

The main problem in this manuscript is that RESULT only listed the analysis results, lacking the support of experimental data. Especially in the "Phenology traits" section, the authors should explain the important conclusions rather than only list all the contents in table 2. Moreover, since there was more than one QTL per chromosome, an additional column of positional information (such as Bt6:8628087) should be added in table 2 to resolve the ambiguity of lines 114-120.

In addition, there were still some problems need to be confirmed in this manuscript.

The chromosome information in table two was replaced with the information on the marker so that rather than Bt6 it reads Bt6:8628087. The results and discussion were combined with more experimental data from other studies being integrated into those paragraphs as citations.

1> In lines 94-95, "of" should be added between "correlation" and "Best" in the sentence "Broad Sense Heritability (Reliability), the mean, standard deviation, min, max and correlation Best Linear Unbiased Estimates (BLUEs) were calculated to ...".

We added "of" in text. (Line 96 in current manuscript)

2> In line 106, the number "-0.51" differed from "-0.53" labeled in Figure 2.

Updated in manuscript to match -0.53 in figure 2.

3> In lines 342-343, missing close parenthesis.

Updated in text

4> In figure 1, I was not sure if changing the order of the chromosomes is better to show the results. I thought the authors should add a ruler or specification related to the orientation of the chromosomes.

We changed the chromosomes to be ordered 1 through 7 and added a ruler specification showing the orientation of the chromosome. Please note that the additional SVG file of updated image in the extra files submitted for the publication.

5> In table 2, the sentence “where trait includes plant height (PH) (FRS), (VPN), or (AWN50)” was incorrect.

This was re-written for readability

6> All the figures should be changed to a vector graph to increase clarity.

Manuscript figures have been remade as scalable vector graphs (SVG). Word cannot take the size of the SVG figures so PNGs were left in text and the SVG files were uploaded to the submission as additional files.

Reviewer #2 (Remarks to the Author):

This manuscript presented WGS and GWAS works for *B. tectorum*. The results are interesting and I recommend to publish this paper as it would provide a pool of valuable genomic data for the people in the weed science to use for subsequent genomic and genetic studies.

Few specific comments:

L127: comma after “In rice (*Oryza sativa* L.)”

Comma has been added in text

L139. Negative correlation between PH and phenology traits was also report by (<https://link.springer.com/article/10.1007/s10681-020-02686-8>)

That citation is highly relevant to our work in ways additional to the negative correlation. The citation was added for the negative correlation in the discussion (Line 145)

L163: First time use this acronym so should spell it out here.

Acronym was spelled out in text

L154-157: do not need to include P values as you put them in the Table already.

P-values were removed from text

L185: while 500 kb was chosen as the interval for searching for candidate genes, in Table 3 it states that the candidate genes are within 250 kb?

A 500Kb interval with 250Kb on either side of the SNP was searched, I further specified in the manuscript that there was a 500Kb window spanning 250Kb to either side of the SNP searched (Line 236).

L187: needs a dot to finish the previous sentence.

The period was added to finish the specified sentence

L195-204: FRS in the paper cited (no.49) is the Far-red response mutant. Do you mistaken it with the RFS acronym? If not, why you mentioned only three genes as potential candidate first and then later mentioned something else could be candidate? Did you blast the sequence to know if FRS5 is near the Bt2?

No, *FRS5* stands for *FAR1-RELATED SEQUENCE 5*, the *FAR1* of “*FAR1-RELATED SEQUENCE 5*” stands for “far-red-impaired response”. This is in the citation (no.59). *FRS5* is from the gene annotation that used blast. *FRS5* was near a QTL associated with first ripe seed (FRS). We specified they were candidate genes for a QTL that has had *DREB1F* on it and they were brought up due to epistasis between *DREB1F* and *GA20OX*. We wrote out *FAR1-RELATED SEQUENCE 5 (FRS5)* in the text to avoid ambiguity. (Lines 250-262)

Is it what you meant that for RFS QTL on Bt1, there are two candidates gene including *FRS5* and *DREB1F*, but *DREB1F* is the more likely candidate?

We meant *FAR1-RELATED SEQUENCE 5 (FRS5)* on *Bt1 (173453655)*, We corrected the manuscript to reflect which QTL the *FRS5* and *DREB1F* were associated with. We discuss how the interaction validates *DREB1F* as being the gene causing the effect of the QTL. Its hard to rule *FRS5* out because it is a photoreceptor but we did provide more evidence for *DREB1F* in the manuscript. (Lines 250-262)

L227: change the citation format here

Citation format fixed in manuscript.

L239: in=is?

Yes, that is what was meant, it has been replaced in manuscript.

L254: you cited an oat paper, but can add barley papers as this gene is important in barley as well, and there are tens of paper about that in barley

The citation you provided about the negative correlation between height and phenology in barley also finds the same gene (*GA20OX2*). We added that citation and another from barley in the manuscript. (Lines 145 & 310).

Overall questions: why PH had a wide variation and a high heritability as well but only 1 QTL detected? It is hard to believe that only 1 QTL control height in *B. tectorum*. Is there any other papers in literature map and identify only 1 QTL for height in weeds? I meant is it common in weeds to have QTL controlling height? Barley has >10 and these two are similar so is there any way to explain this?

We do not believe that only 1 QTL is controlling height in *B. tectorum*. Often, GWAS analyses do not find all of the genes controlling a trait (missing heritability). When there are large effect leading SNP identified, it can be hard to identify the polygenic component of a trait. For example, In Salmon (*Salmo*

salar) it was thought after an initial GWAS study that age at maturity was controlled by one large effect QTL with no polygenic component (<https://doi.org/10.1038/nature16062>) but a subsequent GWAS with a larger population identified the same major QTL but also identified a polygenic component consisting of many genes with small effects (<https://doi.org/10.1186/s12711-020-0529-8>) (Lines 152-164)

The correlation between VPN to FRS and AWN50 was very similar (0.97 to 0.98), however, the number QTL detected for VPN and FRS are 3X more than AWN50. Do you have any explanation?

The GWAS model that we used is a multi-locus model that goes through iterations of model selection thus varying the results based on what SNP are found significant by the model. During the multiple iterations of model selection, the markers that were selected in previous iterations are included with the linear model used for testing and thus selecting the markers to be included in the fixed effect model in BLINK. During model selection in early stages of BLINK even slightly different results in the model selection can lead to entirely different sets of SNP being identified for traits. In corn, a GWAS revealed in corn correlated phenology traits yielded different numbers of significant SNP. (<https://academic.oup.com/pcp/article/61/8/1427/5809534>) (Lines 224-236)

What is the coverage of the sequencing works (is it 40X)?

For assembling the reference genome the coverage was 68.6X. (Line 379)

Reviewer #3 (Remarks to the Author):

Revolinski et al. present a study exploring the *Bromus tectorum* genome and potential genes associated with plasticity in traits that have been recognized as important to the species success as an invasive. The authors present a high-quality genome that will serve as a valuable resource to multiple communities of scientists, especially weed genomicists. The remainder of the study is promising in the highlighting of potential genes that may contribute to traits identified as instrumental in the success of cheatgrass as an invasive.

Overall, I found much of the paper to be straightforward. The introduction seems to be lacking, but that may be partially an artifact of the format of this journal with shorter articles. Below I mention some additional information that would be helpful in the introduction.

There were some portions of the Results that read like Discussion (Lines 129-140 and 188-192). Again, this may be a format of the journal issue. Other articles in the journal do combine these two sections.

The results and discussion sections were combined.

It was unclear what coverage cutoff was used in the GWAS to identify SNPs (Lines 359-362). This would be valuable information to have in the paper.

Although the vcf file was not explicitly filtered for read depth the minimum read depth was 54 with a median read depth of 808 which is well above typical cutoffs. (Lines 429-431)

Figure 3 seems like a supplemental figure. There is not a lot the reader gets from looking at the figure that would not be more easily understood from text.

Figure 3 has been converted to a supplemental figure (figure S2).

I was surprised that flowers/inflorescences were not used in the annotation give how important phenology is to much of the interpretation of the study. There is no report in the main text or the supplement regarding number of genes annotated or additional annotations for other features (like transposons). It would be valuable to have this included in a table somewhere.

While it would have been best to use transcripts of flowers/inflorescences directly from *B. tectorum*, secondary transcriptomes from several related monocot species with samples from different tissues including inflorescence related tissues were used for the annotation. When the genome was assembled and annotated it was not apparent that the first use of the genome would be mapping flowering time genes thus inflorescence tissues were not directly sampled for transcriptomic analysis. A supplemental table has been added that contains information on features such as genes and various repeats including transposons. (Supplementary information)

Line 12: Need a “the” between “outcompete” and “native”.

We have added “The” between “outcompete” and “native” on line 12

Line 19: No “,” needed after “species”.

The comma has been deleted on line 19.

Lines 39-45: It was stated that cheatgrass likely originated in the US through a bottleneck, but this section is discussing genetic diversity of ancestral populations. Are these the original founding populations or farther back in time? Is there evidence of other instances of gene flow from native populations since the original introduction? Some more description and clarity of the known history of invasion and genetic diversity of invasive and native populations would be helpful here.

Bromus tectorum in North America likely originated from a number of different pre-adapted specialist genotypes with more genetic diversity in the Eurasia than North America (<https://www.jstor.org/stable/41416310>) (Lines 36-41)

Lines 45-48: This sentence is a bit unclear. Plasticity is the variable response to a range of factors and not necessarily adaptive. Unless this is meant to say that exhibiting plasticity itself is adaptive here. I think this is an issue of the sentence structure tripping me up.

The plasticity allows for individual plants to survive in-season fluctuations and establish them-selves while the standing genetic variation, originating from all of the genotypes that were able to survive the new environment, allows for these individual plants to make-up populations with high genetic variation making *B. tectorum* resilient to near-term and long-term climactic changes. The sentence was updated in the manuscript to reflect this clarification. (Lines 48-50)

Line 57: North American

Manuscript updated to reflect this change. The n was added.

Line 58: Should be Poaceae.

The manuscript was updated with this correction.

Line 63: Triticeae is a tribe, not subfamily.

The manuscript was updated to read “tribe” instead of subfamily

Line 202: GA needs to be written out.

GA was written out as “gibberellic acid (GA)” because it was the first time the abbreviation was used in the manuscript.

Line 265: This seems like an overstatement. This was one study for one species. This is overgeneralizing what selfing species may do. There are other selfing grasses with phenotype and genomic data sets you compare to see if this is a trend or not.

We realized it was an overstatement once we reviewed the data. Theoretical work does indicate that divergent selection with deleterious mutations results in the emergence of large effect QTL controlling adaptive traits (<https://doi.org/10.1111/nph.16186>) so we used the results to support that theory instead (Lines 321-332).

Line 277: Need a period after reference.

The error has been corrected in the manuscript.

Line 304: Reference issue.

The error has been corrected in the manuscript.

REVIEWERS' COMMENTS:

Reviewer #1 (Remarks to the Author):

The revision well addressed my concerns. I have no further questions or suggestions.

Reviewer #2 (Remarks to the Author):

In my view, the authors have well-addressed all the comments, suggestions and questions. Therefore, I recommend this manuscript to be published.